# The Effects of Preslaughter Electrical Stunning on Serum Cortisol and Meat Quality Parameters of a Slow-Growing Chinese Chicken Breed

**DOI:** 10.3390/ani12202866

**Published:** 2022-10-20

**Authors:** Wenpeng Li, Chao Yan, Kris Descovich, Clive J. C. Phillips, Yongyou Chen, Huijing Huang, Xuan Wu, Jian Liu, Siyu Chen, Xingbo Zhao

**Affiliations:** 1College of Animal Science and Technology, China Agricultural University, Beijing 100193, China; 2Agricultural Genomics Institute at Shenzhen, Chinese Academy of Agricultural Sciences, Shenzhen 518124, China; 3School of Veterinary Science, University of Queensland, Gatton, QLD 4343, Australia; 4Institute of Veterinary Medicine and Animal Sciences, Estonian University of Life Sciences, Kreutzwaldi 1, 51014 Tartu, Estonia; 5Curtin University Sustainability Policy (CUSP) Institute, Curtin University, Perth, WA 6845, Australia; 6Nayong County Agriculture and Rural Bureau, Bijie 553300, China; 7Key Laboratory of Animal Molecular Design and Precise Breeding of Guangdong Higher Education Institutes, School of Life Science and Engineering, Foshan University, Foshan 528225, China

**Keywords:** chicken, welfare, preslaughter stunning, meat quality, cortisol

## Abstract

**Simple Summary:**

In China, although preslaughter stunning is commonly performed in large-scale production systems, this practice is less common in smaller-scale systems. To promote the concept of animal welfare, we selected common local chicken breeds and sampled the meat and blood of chickens with and without stunning. The experimental results verified that stunning before slaughter can improve meat quality, increase flavor and reduce the stress on chickens. It is more respectful of animal welfare and provides a theoretical basis for the promotion of animal welfare concepts in China.

**Abstract:**

Electrical stunning is widely utilized prior to a neck cut to induce unconsciousness in order to improve animal welfare and slaughter efficiency in the broiler production industry. However, slaughter without stunning is still very commonly used in China, in part because there is a belief that stunning reduces meat quality. The purpose of this study was to determine whether the physical (hemorrhages, pH, drip loss, and shear force) and chemical (inosinic monophosphate concentration and reducing sugar content) properties of broiler meat differed between chickens in preslaughter stunning and nonstunned slaughter groups, and whether the groups differed in their levels of cortisol as an indicator of stress. Serum cortisol levels of the nonstunned group were nearly twice as high as those in the stunned group (*p* < 0.05). Several meat quality indicators were better in the stunned group than in the nonstunned group. We concluded that electrical stunning prior to slaughter significantly decreases the stress caused by slaughter, resulting in both improved animal welfare and meat quality traits.

## 1. Introduction

Animal welfare has received an increasing amount of attention in China, particularly in the field of animal farming [1]. Among the challenges for animal welfare within food chains, the slaughter method is widely recognized as a determinant of animal suffering and attracts significant attention from industry groups and consumers [2]. Preslaughter stunning of animals is applied to achieve loss of consciousness prior to slaughter being carried out [3]. This is generally considered better for animal welfare than slaughter without stunning [3]. Common stunning methods for livestock include mechanical, gas, and electrical stunning [4,5]. Mechanical stunning is performed on an individual animal using a penetrating or percussive device such as a captive bolt [6]. This is primarily used for large animals such as cattle, sheep, and goats [7]. For the slaughter of large numbers of small animals, such as broilers, this method is generally not efficient. Gas stunning is an alternative to mechanical stunning. The most common gases used to stun broilers are carbon dioxide, nitrogen, and argon. Gas stunning results in lower levels of carcass damage; however, it is expensive and more challenging to perform compared with other methods [4]. For these reasons, gas stunning is not commonly used in broiler production.

Currently, in China, large-scale farms typically slaughter broilers after stunning by delivering an electric shock via a water bath, followed by exsanguination. Electrical stunning (ES) directly applies a current through the animal. When the current is of sufficiently high frequency, it results in a grand mal seizure and irregular ventricular fibrillation, rapidly leading to a loss of consciousness and/or death [8]. Previous research has indicated that a high-frequency current can quickly and effectively induce cardiac arrest and loss of consciousness so slaughter can be humanely carried out [9,10]. In addition to the animal welfare outcomes, ES also has the advantages of being practical to apply and low-cost [11]. However, ES is also thought to cause carcass defects and reduce bloodletting efficiency, with the effects being related to high current frequencies [12]. While reducing current intensity and voltage may reduce carcass defects, this may also decrease stunning efficiency, resulting in negative impacts on animal welfare [9,10,13]. However, the scientific literature has equivocal results. Several studies have shown the effects of different voltage/current combinations on the meat quality of broilers [14,15]. Similarly, different stunning frequencies and electrical current waveforms have different effects on meat quality parameters and welfare [16]. However, others have concluded that meat quality attributes are not affected by the stunning conditions [17]. Regardless of the impact on meat quality, inappropriate combinations of voltage and frequency can lead to an improper death or a return to consciousness before slaughter can be carried out [15]. Thus, for good animal welfare, it is critical that the ES system is sufficient to ensure that the chickens will not regain consciousness or suffer during the slaughter process.

In China, although stunning is generally carried out before slaughter in large-scale production systems [1], the practice is not common in smaller-scale systems. According to industry focus groups, there is a belief that “stunning is harmful to the meat quality” in the southern regions of China [18]. In contrast, in the north, the consensus amongst participants is that stunning does not affect the taste, and the primary reason for not stunning is a lack of understanding of the technology and equipment involved [18]. Therefore, to increase the uptake of stunning in China, both quality concerns and technical expertise should be addressed and evaluated. Several studies have investigated the differences in carcass quality between broilers slaughtered with and without preslaughter stunning [3,19], but they must acknowledge the major differences in the carcass quality of different broiler breeds [20]. The Wumeng black bone chicken is a popular local chicken breed in the south of China [21]: the average weight of adult males is 2650 g and that of hens is 1840 g. Considering its widespread use in the industry, we chose this breed as our experimental material. 

In this study, we aimed to investigate the effects of preslaughter ES on the welfare and carcass quality of Wumeng black bone chicken based on the physical and chemical properties of the broiler meat and serum. 

## 2. Materials and Methods

### 2.1. Animals

The experimental and animal care protocols for this study were approved by the China Agricultural University Laboratory Animal Welfare and Animal Experimental Ethical Inspection (approval number: AW62902202-1-1). This study was performed in Nayong County, located in Bijie, Guizhou, China. A total of 40 180-day-old female Wumeng black bone chickens were used, sourced from the Yuansheng Animal Husbandry Co., Ltd., Bijie, China. This chicken breed is a slow-growing line, with a growth rate of about 11 g/day [22]. Birds were initially reared within a cage system until 70 days of age before being moved to a free-range system. One hundred birds were reared in a single group with approximately 2 m^2^ per bird. The surface of the rearing area consisted of a soil substrate with grass. Birds around 180 days of age are popular with Chinese consumers. Thus, the birds used in this study were chosen to be typically representative of this system. 

### 2.2. Animal Slaughter and Sample Collection

In total, 40 180-day-old birds were randomly selected from 232 birds for sale and allocated into two groups: slaughter without stunning (*n* = 20) and slaughter after ES (*n* = 20). For the stunned group (Treatment S), birds were hung upside down on a chain track fitted with leg shackles, which moved them to a charged water bath where their heads entered the water. A medium voltage, low-frequency alternating current (50 V (48-52 mA), 50 Hz) was applied for 10 s, which is known to induce a loss of consciousness when visually assessed [16]. While the chicken was unconscious, a cut was manually made to the left artery in the lower jaw of the neck in the featherless area of the chicken’s ear, 1.5–2 cm from the head gland and 1.5 cm from the end of the hyoid bone, with subsequent complete exsanguination. The remaining 20 individuals were identically slaughtered; however, they did not receive preslaughter stunning and were therefore slaughtered while conscious (Treatment NS). We did not slaughter NS birds to conduct the experiment. Instead, we collected samples with the consent of the customers who purchased the NS birds. Before slaughter, all birds were kept in the same conditions. All the NS and S birds were slaughtered at the same time. All slaughter procedures were carried out by the same butcher with two years of work experience who was blind to the aim of this study. Blood samples from the artery cut during the slaughter procedure were collected in 9 mL vacuum blood collection tubes (VACUETTE, Greiner Bio-One, Solingen, Germany) to measure cortisol levels. To determine the physical properties of the meat, the left side of both breast and thigh muscles was collected, and fresh samples were analyzed within 24 h after slaughter. In preparation for analysis, the left breast and thigh muscles were divided into 5 sections, each with an approximate weight of 100 g, to investigate physical properties (hemorrhage, pH, drip loss, cooking loss, and shear force). The right sides of the breast and thigh muscles were frozen at −20 ℃ to analyze their chemical properties. They were each divided into 2 equal sections for chemical analysis (inosine monophosphate concentration and reducing sugar content). All analyses were performed in triplicate except for shear force analysis, for which six replicates were used.

### 2.3. Physical Properties of the Meat

#### 2.3.1. Hemorrhage

Hemorrhaging in the breast and thigh muscles was quantified using a visual grading system [23]. Classification of the severity of hemorrhaging was independently performed by an observer trained in the visual grading system and with two years of slaughter experience. Hemorrhaging was classified on a three-point scale, with class 1 representing hemorrhage-free muscles, class 2 representing slight hemorrhage, and class 3 representing muscles with numerous and/or severe hemorrhages. 

#### 2.3.2. pH

Measurements of pH values were performed using a portable pH probe (PH-STAR, Matthaus, Germany) by inserting the probe into the center of the muscle sample within 24 h of slaughter. The pH meter was calibrated prior to the measurement using two buffer solutions at pH 4.0 and 7.0. The probe was inserted into the muscle three times (in different locations), and the final pH value used was the mean of the three measurements. 

#### 2.3.3. Drip Loss

To determine drip loss, meat samples were weighed before analysis. Each sample was suspended in a sealed polyethylene bag with an aluminum wire hook and sewing thread. The sealed bag was filled with nitrogen to minimize the contact between the meat sample and the bag. Both were then suspended in a refrigerator at 4 ℃. After storage for 24 h, the surface water of the meat was gently dried with filter paper before weighing. The drip loss values were calculated using the difference between the initial and final weights of the samples, and the results are expressed as a percentage of the initial weight [24]. 

#### 2.3.4. Cooking Loss

To determine cooking loss, samples were preweighed and subsequently placed in sealed polyethylene bags. The bags were then transferred into continuously heated boiling water until the samples reached an internal temperature of 75 ℃, as measured by a portable thermometer (Hengko, Shenzhen, China). The bags were then removed from the boiling water and cooled to 25 ℃, as determined by the portable thermometer, at which point surface water was gently wiped from the sample using a paper towel. Cooking loss values were calculated as the difference between the initial and final weights of the samples, and the results are expressed as a percentage of the initial weight [25].

#### 2.3.5. Shear Force

To measure shear force, the same samples from the cooking loss measurements were used. They were placed overnight at 4 ℃ in a fridge (BCD-251U, Hisense, Qingdao, China) and then removed and cut into 1 ✕ 1 ✕ 4 cm long strips to determine the shear force using a muscle tenderness meter (C-LM3B, Tenovo, Beijing, China) by the Warner-Bratzer method [25]. The values collected were a mean of six measurements expressed in kilogram-force [25].

### 2.4. Chemical Properties of the Meat

#### 2.4.1. Cortisol Assays

Blood samples were allowed to stand for 30 min at room temperature, following which they were centrifuged for 10 min at 1500× *g*. The separated serum was collected and stored at −20 °C until further analysis. Cortisol concentrations were determined from the serum with a cortisol ELISA kit (Cayman Chemical, Ann Arbor, MI, USA) used in accordance with the manufacturer’s instructions. An enzyme-labeled instrument (Multiskan FC, Thermo Scientific, Waltham, MA, USA) was used to measure the absorbance at 420 nm of the samples. A standard curve with an R^2^ value of 0.9914 was generated to determine the amount of cortisol in an unknown sample. The samples were analyzed in triplicate, and values used for data analysis were the mean of the three values.

#### 2.4.2. Inosine Monophosphate Concentration

The inosine monophosphate (IMP) concentration was determined with a chicken inosine monophosphate ELISA kit (LaiEr bio, Hefei, China) used in accordance with the manufacturer’s instructions. An enzyme-labeled instrument (Multiskan FC, Thermo Scientific, Germany) was used to determine optical density at 450 nm. A standard curve with an R^2^ value of 0.9936 was generated to determine the amount of inosine monophosphate. The samples were analyzed in triplicate, and the values used for data analysis were the mean of the three values.

#### 2.4.3. Reducing Sugar Content

The reducing sugar (RS) content was determined with a reducing sugar ELISA kit (Solarbio, Beijing, China) used in accordance with the manufacturer’s instructions, and the results are expressed as a percentage of the sample weight. A UV spectrophotometer (UV-5100, METASH, Shanghai, China) was used to determine optical density at 540 nm. A standard curve with an R^2^ value of 0.9919 was generated to determine the amount of reducing sugar. The samples were analyzed in triplicate, and values used for data analysis were the mean of the three. 

### 2.5. Statistical Analysis 

The data of hemorrhage scores were not continuous, so we used the chi-squared test to judge the significance of the difference between the two groups, and the other data were all analyzed using Student’s t-test. *P* < 0.05 was classified as indicating a significant difference. All analyses were performed using SPSS 20.0 software (IBM, Chicago, IL, USA). The experimental data are presented as the mean ± standard error (SEM).

## 3. Results

### 3.1. Physical Properties of the Meat

The hemorrhage scores from the breast muscle were significantly higher in chickens slaughtered without stunning (*p* < 0.05) (Table 1). No severe hemorrhages (Score 3) were found in the S carcasses, but the breast muscle from two broilers slaughtered without stunning displayed severe bruising. The pH values measured from the breast muscle of unstunned birds were significantly lower than those from stunned birds (*p* < 0.05) (Table 1). Unstunned birds had a greater drip loss for both breast and thigh meat than the stunned birds (Table 1). Cooking loss was greater in the unstunned than stunned birds for breast muscle (*p* < 0.05), but there was no difference between groups in the thigh muscle. The shear force values of meat from unstunned birds were greater than those for meat from unstunned birds, for both breast and thigh muscles (*p* < 0.05). 

### 3.2. Chemical Properties of the Blood and Meat

The cortisol levels measured in the blood of stunned chickens were considerably lower than those of unstunned chickens (*p* < 0.05). The inosine monophosphate content measured from the breast muscles of stunned chickens was greater than that from unstunned chickens (*p* < 0.05), while the inosine monophosphate concentration of the thigh muscles was unaffected by the experimental treatment. Analysis of pooled data revealed that in both breast and thigh muscles, the reducing sugar content in the chicken was similar in the two groups (Table 2).

## 4. Discussion

The results from our study indicated that preslaughter stunning results in lower cortisol levels and better meat quality based on water-holding capacity and shear force. The physical properties of the resulting meat were more affected by preslaughter stunning than the chemical properties, based on the number of measures that were statistically significant.

### 4.1. Physical Properties of the Meat

The general belief is that ES is more respectful of animal welfare [14]. However, it has also been argued that ES causes contraction, movement between muscles, and abnormal positioning of the animal during the stunning procedure, resulting in rupturing of blood vessels and damage to muscle fibers [15]. In the present study, the hemorrhage scores of breast muscle from stunned birds were significantly lower (better) than those of unstunned birds. This indicated that stunning at the voltage/current used has a beneficial effect on the resulting carcass quality. Our results contradict those of previous studies, which suggested that ES causes more severe bruising [12,15]. The reason for this difference may be that we compared two types of slaughter (with and without ES prior to slaughter), while the other studies compared two methods of preslaughter stunning (ES and gas stunning). 

pH values were lower in the unstunned group than in the stunned group, especially for the breast muscle (*p* < 0.05). This may be explained by anaerobic glycolysis during slaughter in the unstunned birds. When an animal is slaughtered, the oxygen supply chain is cut off, and anaerobic glycolysis becomes the main metabolic pathway for energy production [26]. The pH value of the muscle in the early postmortem period, and the final pH value greatly affect meat quality after slaughter. In the early postmortem period, if the pH value of the muscle is low and the carcass temperature high, degeneration of myofibril protein and sarcoplasmic protein in the muscle occurs, increasing the incidence of PSE-like (pale, soft, and exudative) meat in chicken. If the pH value slowly declines in the early postmortem period, it results in cold contraction, thereby reducing the water retention and tenderness of the muscle [12]. For chickens that are conscious at the slaughter stage, more lactic acid likely accumulates in the muscles than in stunned chickens; our observations suggested that this occurred because the unstunned birds extensively flapped their wings during slaughter (although behavior was not formally recorded during this study). This may explain why the pH values of the unstunned group were lower than those of stunned group. In line with our results, previous studies have indicated that stunning before slaughter reduces the flapping of broilers’ wings during the slaughter process, thereby reducing the speed of glycolysis after slaughter [12,27]. Moreover, the decrease in pH value is likely to have been caused by the increase in flapping wings induced by stress and struggle. In stressful situations, the secretion of catecholamines and glucocorticoids stimulates hepatic glycogenolysis, leading to elevated blood glucose levels [28,29]. After exsanguination, the muscle becomes hypoxic and triggers anaerobic glycolysis, in which glycogen is hydrolyzed to lactic acid. As a result, the pH of the meat drops [30]. Therefore, we propose that pH can be used as a reference for assessing stress, with lower pH levels indicating higher levels of stress during slaughter.

In our study, drip loss from breast and thigh meat was greater in unstunned birds. Drip loss refers to the loss of liquid from the muscle using only the action of gravity over a period of time [31]. pH levels in meat are known to impact drip loss, as lower pH levels cause myofibril contraction, which makes the voids in the myofilament smaller and water loss higher [32,33]. 

Cooking losses are related to a reduction in meat quality, and the index is one of the most commonly used to describe the water retention of processed meat [34]. Cooking loss is affected by several factors, including the fat content, protein denaturation under heating, and the volume lost during thawing [35]. In the current study, a significant difference was found in breast muscle cooking loss between the experimental groups. Both preslaughter factors and postmortem treatments are related to changes in meat quality traits, but these changes ultimately often affect muscle water-holding capacity by affecting the proteins that comprise the muscle [36]. Again, a lower pH value may cause degeneration of the myofibril protein and sarcoplasmic protein in the muscle [12,32,33], and the resulting protein denaturation can lead the myofibril mesh and muscle cells to stretch, directly affecting the muscle water-holding capacity after slaughter [37]. Therefore, the lower cooking loss of the muscle from stunned birds may be associated with a more rapidly decreasing pH value in the early stages after slaughter. 

Shear force value is an indicator of muscle tenderness, which is a key factor determining the quality of meat [38]. The main variables that affect meat tenderness include sarcomere length, connective tissue content, and the degree of degradation of myofibril protein [39]. Long sarcomeres result in more tender meat, and sarcomere length is mainly affected by rigor mortis and improper slaughter techniques [40,41]. In the current study, myofibril contraction was likely caused by the lower pH value in the unstunned group promoting the degradation of the myofibril protein [33]. As the lower pH value may have resulted from struggling in the unstunned birds, this may explain the higher shear force value of unstunned birds’ meat compared with that of meat from stunned birds.

### 4.2. Chemical Properties of the Meat

Inosine monophosphate can improve the meat flavor of livestock and poultry [42,43] and is internationally used as an important index for measuring meat flavor [44]. Because IMP has weak chemical bonds, such as glycosidic and ester bonds, its stability depends on temperature and pH [45]. This may explain why, in our study, IMP concentration was found to be higher in the breast muscle of the stunned birds, with the higher initial pH in the breast muscle of the unstunned group apparently induced by wing flapping and anaerobic glycolysis. The lower pH may have resulted in higher degradation of IMP in the unstunned chickens. As we did not formally investigate the behavior of the birds, this is only a hypothesis; however, it may explain why there was a significant difference in inosine monophosphate concentration in the breast muscle but not the thigh muscle.

The hypothalamic–pituitary–adrenocortical (HPA) axis is a critical adaptive system that maximizes survival potential in the face of physical or psychological challenges [46]. The activated hypothalamic corticotropin-releasing hormone neurons stimulate the pituitary gland to release corticotropin, which in turn stimulates the adrenal cortex to secrete cortisol [47]. Cortisol concentration is a widely used indicator for the assessment of acute stress [48]. In the current study, slaughtering the conscious birds without stunning appears to have been a major stressor, resulting in activation of the HPA axis and increasing the release of cortisol [49]. 

The Maillard reaction, a chemical reaction between reducing sugars (e.g., glucose and ribose) and free amino acids, generates many important flavor compounds and thereby intensifies the taste of cooked meat [50]. Glucose, as a type of reducing sugar, is expected to have been utilized in large quantities during slaughter in the unstunned birds because we observed that they reacted behaviorally (e.g., wing flapping and struggling). However, in the current study, no differences in reducing sugar content were found between the two experimental groups. One of the main reducing sugars in meat is ribose [51]. Ribose is one of inosine monophosphate degradation products, and the higher inosine monophosphate degradation rate of unstunned birds induced by lower pH may have resulted in the production of more ribose [52]. 

Overall, the results of this study suggest that in comparison with the traditional broiler slaughter method, preslaughter ES achieves a better product quality, which aligns with the results of previous research [3]. In addition, the lower cortisol concentrations and higher pH values of the stunned group compared with those of the traditional slaughter group indicated that preslaughter ES results in better animal welfare [12,27,53]. Therefore, preslaughter stunning is not only important for meat quality but also contributes to improved welfare for the chickens and should therefore be more widely adopted.

## 5. Conclusions

Electrical stunning prior to slaughter improved the physical properties of meat, including pH, drip loss, cooking loss, and shear force. In addition, inosine monophosphate content, an important indicator of the umami taste of meat, was also increased. In combination with a decrease in serum cortisol levels, these results suggest that preslaughter stunning can significantly both improve animal welfare and product quality. 

## Figures and Tables

**Table 1 animals-12-02866-t001:** The physical properties of breast and thigh muscles of broiler chickens.

Muscle Type	Slaughter Method (Mean ± SEM)		*p*-Value
	Stunning (S)	No Stunning (NS)	
Breast			
Hemorrhage score (1–3)	1.15 ± 0.08	1.60 ± 1.15	0.013
pH	5.95 ± 0.07	5.67 ± 0.04	<0.001
Drip loss (%)	3.21 ± 1.12	4.11 ± 1.11	0.002
Cooking loss (%)	26.34 ± 1.26	29.48 ± 1.38	0.017
Shear force (kgf)	22.69 ± 1.44	27.63 ± 1.28	0.012
Thigh			
Hemorrhage score (1–3)	1.15 ± 0.08	1.35 ± 0.11	0.152
pH	6.00 ± 0.03	5.88 ± 0.05	0.096
Drip loss (%)	2.32 ± 1.45	3.11 ± 1.36	0.012
Cooking loss (%)	32.16 ± 1.35	32.24 ± 1.68	0.301
Shear force (kgf)	30.42 ± 1.38	41.81 ± 1.65	<0.001

**Table 2 animals-12-02866-t002:** Chemical properties of breast and thigh muscles of broiler chickens in the stunned (Treatment S) and unstunned (Treatment NS) groups.

Compounds	Slaughter Method (Mean ± SEM)		*p*-Value
	Stunning (S)	No Stunning (NS)	
Cortisol (pg/mL)	273.51 ± 40.99	504.29 ± 73.42	0.003
Breast			
IMP (nmol/L)	214.05 ± 5.43	179.97 ± 5.80	<0.001
Reducing sugar (μg/g)	545.80 ± 61.01	519.12 ± 63.53	0.698
Thigh			
IMP (nmol/L)	215.09 ± 5.53	201.18 ± 6.47	0.056
Reducing sugar (μg/g)	402.38 ± 45.52	408.94 ± 61.61	0.174

IMP = inosine monophosphate, RS = reducing sugars. *p*-values are derived from t-tests.

## Data Availability

Data is available on request from the Corresponding Author.

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
