# Peer review of "The Effects of Preslaughter Electrical Stunning on Serum Cortisol and Meat Quality Parameters of a Slow-Growing Chinese Chicken Breed"

_animals, 2022, doi:10.3390/ani12202866_

Round 1
Reviewer 1 Report
The study evaluates the effects of two different slaughtering methods (slaughter with previous electrical stunning and slaughter without stunning) on serum cortisol and meat quality parameters (both physical and chemical) in a common Chinese chicken breed.
The results show that stunning before slaughter can improve meat quality, increase flavour substances and reduce the stress of chickens providing a theoretical basis for the promotion of animal welfare concepts in China.
The manuscript is well structured and it is attractive for readers not only in the field but also for industry groups and consumers.
To make comprehension and reading easier, however, authors should explain the acronyms putting them after the long form and decide which form (the long or the short form) of the group names to use in the rest of the manuscript; for example in the lines 55 and 61 the acronym ES stands for electronic stunning?
Furthermore, few typographical errors can be found in the text:
Line 128: “Hemorrhage” is missing amongst the physical properties in the blankets.
Line 178: after “was used” the phrase could be continued as in the line 185 with “for determine optical density” ?
Lines 21;295;296;314: instead of “flavor” it could be better use “flavour”.
The manuscript is scientifically sound and the experimental design appropriate to test the hypothesis formulated.
The manuscript’s results are reproducible based on the details given in the “Materials and methods” section. However, it is not clear how the 40 chickens initially examined have been selected from the free range for being parts of the two slaughtering groups (line 108).
The tables are appropriate and easy to interpret but there are some typographical error:
Line 212: according to Table 1 cooking loss was greater in the NS chickens than in the S birds for breast muscle but according to the line 212 (Paragraph 3.1) is the opposite. Maybe the line 212 contains a simple typographical error?
Line 216: use the capital letter for the word “the” in the Table 1 caption
Line 225: use the capital letter for the words “chemical” and “treatment S” in the Table 2 caption.
The cited references are mostly recent publications and relevant. For greater clarity in the Discussion (Paragraph 4.1) could be improved as follow.
Lines 243-245: The concept could be expressed more clearly, in the first case it is a comparison between two types of slaughter (with and without stunning) in the second case it is a comparison between two methods of pre-slaughter stunning.
Lines 250-254: For a more complete and clear explanation, authors could be referred to
Barrasso, R.; Ceci, E.; Tufarelli, V.; Casalino, G.; Luposella, F.; Fustinoni, F.; Dimuccio, M.M.; Bozzo G. Religious slaughtering: Implications on pH and temperature of bovine carcasses. Saudi J Biol Sci. 2022, 29(4), 2396–2401. doi: 10.1016/j.sjbs.2021.12.002.
This study highlighted the influence of different slaughtering methods on the variations of pH and temperature. It demonstrated also that in slaughtering carried out without prior stunning, like in NS broilers examined, there is a negative correlations between the stress level during pre-slaughter and meat physical parameters (such as water holding capacity).
Lines 260-266: The decrease in pH, greater in unstunned chickens (Treatment NS) than in chickens stunned, is due non only to lactic acid accumulation, caused probably by flapped wings, but is also attributable to the higher cortisol values found in this group of broilers. Cortisol released in response to stress in fact, directly stimulates glycogen mobilisation, thus contributing to meat acidification. To highlight the connection between pH variations, serum cortisol levels and meat quality parameters authors could refer to:
Bozzo, G.; Barrasso, R.; Marchetti, P.; Roma, R.; Samoilis, G.; Tantillo, G.; Ceci, E. Analysis of Stress Indicators for Evaluation of Animal Welfare and Meat Quality in Traditional and Jewish Slaughtering. Animals 2018, 8, 43. https://doi.org/10.3390/ani8040043.
The results obtained provide an advancement of the current knowledge on broiler slaughtering making available, for the first time, data about the Wumeng black bone chicken, a popular local chicken breed of the south of China, widely used in the industry.
English language is appropriate enough and understandable; for this purpose in the line 235 could be better using expression such as “electrical stunning is more respectful of animal welfare” instead of the adjective "humane" (also used in the lines 22).
The conclusions are consistent with the evidence and arguments presented.
Taking all this into account the work fits the journal aims.
Author Response
Response to Reviewer 1 Comments
Point 1: Line 128: “Hemorrhage” is missing amongst the physical properties in the blankets.
Response 1: Added
Point 2: Line 178: after “was used” the phrase could be continued as in the line 185 with “for determine optical density” ?
Response 2: Added
Point 3: Lines 21;295;296;314: instead of “flavor” it could be better use “flavour”.
Response 3: Done
Point 4: The manuscript’s results are reproducible based on the details given in the “Materials and methods” section. However, it is not clear how the 40 chickens initially examined have been selected from the free range for being parts of the two slaughtering groups (line 108).
Response 4: Done
Point 5: Line 212: according to Table 1 cooking loss was greater in the NS chickens than in the S birds for breast muscle but according to the line 212 (Paragraph 3.1) is the opposite. Maybe the line 212 contains a simple typographical error?
Response 5: Sorry, I mistaking cooking loss for the strength of water retention. It’s already amended.
Point 6: Line 216: use the capital letter for the word “the” in the Table 1 caption
Response 6: Done
Point 7: Line 225: use the capital letter for the words “chemical” and “treatment S” in the Table 2 caption.
Response 7: Done
Point 8: Lines 243-245: The concept could be expressed more clearly, in the first case it is a comparison between two types of slaughter (with and without stunning) in the second case it is a comparison between two methods of pre-slaughter stunning.
Response 8: Done
Point 9: Lines 250-254: For a more complete and clear explanation, authors could be referred to
Barrasso, R.; Ceci, E.; Tufarelli, V.; Casalino, G.; Luposella, F.; Fustinoni, F.; Dimuccio, M.M.; Bozzo G. Religious slaughtering: Implications on pH and temperature of bovine carcasses. Saudi J Biol Sci. 2022, 29(4), 2396–2401. doi: 10.1016/j.sjbs.2021.12.002.
Response 9: Thanks!
Point 10: Lines 260-266: The decrease in pH, greater in unstunned chickens (Treatment NS) than in chickens stunned, is due non only to lactic acid accumulation, caused probably by flapped wings, but is also attributable to the higher cortisol values found in this group of broilers. Cortisol released in response to stress in fact, directly stimulates glycogen mobilisation, thus contributing to meat acidification. To highlight the connection between pH variations, serum cortisol levels and meat quality parameters authors could refer to:
Bozzo, G.; Barrasso, R.; Marchetti, P.; Roma, R.; Samoilis, G.; Tantillo, G.; Ceci, E. Analysis of Stress Indicators for Evaluation of Animal Welfare and Meat Quality in Traditional and Jewish Slaughtering. Animals 2018, 8, 43. https://doi.org/10.3390/ani8040043.
Response 10: Thanks, it’s very useful!
Point 11: English language is appropriate enough and understandable; for this purpose in the line 235 could be better using expression such as “electrical stunning is more respectful of animal welfare” instead of the adjective "humane" (also used in the lines 22).
Response 11: Done
Reviewer 2 Report
General comments and major concern
The study is of some interest however, the ethical assumption of the study is questionable. I can understand that in some not authorized situation stunning is not applied, but this should not be an issue. In the era of animal welfare in which stunning is requested to guarantee animal welfare during slaughtering procedures, it is not clear which is the interest to study the difference between stunning and no stunning birds before slaughtering. I could understand an experimental design in which different pre-slaughter stunning are applied.
A further concer is related to the no-stunned birds, that have been presumably slaughtered in different condition and by different person.
Minor remarks
Abstract
L 26: remove ‘and many other countries’ or specify in which Countries
LL 31-35 provide some numerical data and P values
Introduction
L 40 replace ‘agriculture’ with ‘farming’
LL55-56 ‘Electrical stunning (ES)’ introduce (ES) to be used in the f
ollowing lines
L 62 ES instead of electrical stunning; correct throughout the manuscript. Once you introduce an abbreviation, please use it.
LL 86-89 provide more phenotypic detail about thus chicken (also adult body weigh) and if possible provide a picture
Mat & Method (major concern)
LL 119-121 the slaughtering procedure were performed in different context and by different operators, so the experimental design does not resulted balanced
Results and Discussion are well presented
Author Response
Response to Reviewer 2 Comments
Point 1: L 26: remove ‘and many other countries’ or specify in which Countries
Response 1: Done
Point 2: LL 31-35 provide some numerical data and P values
Response 2: Done
Point 3: L 40 replace ‘agriculture’ with ‘farming’
Response 3: Done
Point 4: LL55-56 ‘Electrical stunning (ES)’ introduce (ES) to be used in the following lines
Response 4: Done
Point 5: LL 86-89 provide more phenotypic detail about thus chicken (also adult body weigh) and if possible provide a picture
Response 5: Done, but the picture is not available because we do not have permission to enter the farm's breeding grounds.
Point 6: LL 119-121 the slaughtering procedure were performed in different context and by different operators, so the experimental design does not resulted balanced
Response 6: Sorry, we didn't express clearly about the slaughter of NS chickens. First, the night before the sale, broilers are initially gathered together, so the condition before slaughter is the same. When there is an online order or a larger order, the corresponding number of chickens will be slaughtered after stunning, and then packaged and shipped. And when individual orders are received in small quantities, non-stun slaughter is used. Under this sales model, we believe that the various variables of noise or systematic error have been minimized under the condition of ensuring animal welfare. Therefore, all the NS and S birds were slaughtered at the same time. All slaughter procedures were carried out by the same butcher with two years of work experience who were blind to the aim of this study.
Reviewer 3 Report
ID 1906031: The Effects of Pre-Slaughter Electrical Stunning on Serum Cortisol and Meat Quality Parameters of a Slow-Growing Chinese Chicken Breed.
The manuscript is well written and easy to read. However, there are some important elements that call into question whether or not it should be published in Animals.
Line 92: Hypotesis: “We hypothesised that electrical stunning provides benefits for animal welfare and hence meat quality”.
This hypothesis has already been verified in previous studies, even carried out in chickens. It has already been established that pre-slaughter stunning reduces stress and does not negatively affect meat quality, compared to when birds are slaughtered without electrical stunning. Based on this, I consider that the research does not have a significant scientific contribution. Why would one think that this prior knowledge is different in another chickens breed?.
Materials and Methods Section
Lines 119-120: “We did not slaughter NS birds in order to conduct the experiment. Instead, we collected samples with consent of the customers who purchased the NS birds”.
This sentence suggests that conditions and factors that could affect the response variables in both groups were not controlled. Apparently, they only had control over the electrically stunned group. So, under this condition of the analyzed samples of the NS group, there is little or no validity of the information, so there is no control of various variables that can cause noise or systematic error. Therefore, the comparison between both experimental groups does not have statistical validity.
Line 201. Statistic Analysis section:
“The experimental data were presented as the mean ± standard error (SEM) of three measurements except for shear force which was from a mean of six measurements.”.
As established in the experimental animals section (2.1), it mentions that 20 chickens per group were used. So the results in the tables should be the mean of n=20. Was this not so?
Results Section
Table 1. The reported values of the shear force I think are incorrect. Possibly instead of units of force they are Newtons. 40 Kg F is too much.
Author Response
Response to Reviewer 3 Comments
Point 1: Line 92: Hypotesis: “We hypothesised that electrical stunning provides benefits for animal welfare and hence meat quality”.
This hypothesis has already been verified in previous studies, even carried out in chickens. It has already been established that pre-slaughter stunning reduces stress and does not negatively affect meat quality, compared to when birds are slaughtered without electrical stunning. Based on this, I consider that the research does not have a significant scientific contribution. Why would one think that this prior knowledge is different in another chickens breed?
Response 1: You are right, relevant studies have already proved the advantages of pre-slaughter stunning, so we removed the sentence. However, we have discussed the relationship between meat quality and stress levels. Moreover, we also verified the concentration changes of inosinic acid and reducing sugar under lower stress conditions, which play important roles in the flavor of meat. We also discussed the relationship between stress and flavour substances to clarify the importance of lower stress level. We believe that this is different from prior knowledge and can promote animal welfare more comprehensively and effectively.
Point 2: Lines 119-120: “We did not slaughter NS birds in order to conduct the experiment. Instead, we collected samples with consent of the customers who purchased the NS birds”.
This sentence suggests that conditions and factors that could affect the response variables in both groups were not controlled. Apparently, they only had control over the electrically stunned group. So, under this condition of the analyzed samples of the NS group, there is little or no validity of the information, so there is no control of various variables that can cause noise or systematic error. Therefore, the comparison between both experimental groups does not have statistical validity.
Response 2: Sorry, we didn't express clearly about the slaughter of NS chickens. First, the night before the sale, broilers are initially gathered together, so the condition before slaughter is the same. When there is an online order or a larger order, the corresponding number of chickens will be slaughtered after stunning, and then packaged and shipped. And when individual orders are received in small quantities, non-stun slaughter is used. Under this sales model, we believe that the various variables of noise or systematic error have been minimized under the condition of ensuring animal welfare. Therefore, all the NS and S birds were slaughtered at the same time. All slaughter procedures were carried out by the same butcher with two years of work experience who were blind to the aim of this study.
Point 3: Line 201. Statistic Analysis section:
“The experimental data were presented as the mean ± standard error (SEM) of three measurements except for shear force which was from a mean of six measurements.”.
As established in the experimental animals section (2.1), it mentions that 20 chickens per group were used. So the results in the tables should be the mean of n=20. Was this not so?
Response 3: Sorry, this is a misrepresentation, the results were the mean of 20 birds.
Point 4: Results Section
Table 1. The reported values of the shear force I think are incorrect. Possibly instead of units of force they are Newtons. 40 Kg F is too much.
Response 4: Because Wumeng black bone chickens are mainly produced in plateau areas (the average altitude is above 1600m) and mostly free-ranged in mountains, range of activities far beyond most chicken breeds in China, this breed is not famous for the tenderness of the meat. Althogh there is no evidence that the tenderness is affected by these two factors, the high shear force of Wumeng black bone chicken is normal. Therefore, in order to avoid the inconvenience caused by the large value, we choose kgf to display the results instead of Newtons.
Round 2
Reviewer 2 Report
The author improved the manuscript according the reviewer's suggestions
Reviewer 3 Report
The observations were addressed in this new version.
Thanks